# Mild Heat Stress Alters the Physical State and Structure of Membranes in Triacylglycerol-Deficient Fission Yeast, *Schizosaccharomyces pombe*

**DOI:** 10.3390/cells13181543

**Published:** 2024-09-13

**Authors:** Péter Gudmann, Imre Gombos, Mária Péter, Gábor Balogh, Zsolt Török, László Vígh, Attila Glatz

**Affiliations:** 1Biological Research Centre, Institute of Biochemistry, HUN-REN, 6726 Szeged, Hungary; gudmann.peter@brc.hu (P.G.); gombos.imre@brc.hu (I.G.); peter.maria@brc.hu (M.P.); balogh.gabor@brc.hu (G.B.); torok.zsolt@brc.hu (Z.T.); vigh.laszlo@brc.hu (L.V.); 2Doctoral School of Environmental Sciences, University of Szeged, 6720 Szeged, Hungary

**Keywords:** triacylglycerol, diacylglycerol, fission yeast, lipid droplet, heat shock, organelles, membranes

## Abstract

We investigated whether the elimination of two major enzymes responsible for triacylglycerol synthesis altered the structure and physical state of organelle membranes under mild heat shock conditions in the fission yeast, *Schizosaccharomyces pombe*. Our study revealed that key intracellular membrane structures, lipid droplets, vacuoles, the mitochondrial network, and the cortical endoplasmic reticulum were all affected in mutant fission yeast cells under mild heat shock but not under normal growth conditions. We also obtained direct evidence that triacylglycerol-deficient cells were less capable than wild-type cells of adjusting their membrane physical properties during thermal stress. The production of thermoprotective molecules, such as HSP16 and trehalose, was reduced in the mutant strain. These findings suggest that an intact system of triacylglycerol metabolism significantly contributes to membrane protection during heat stress.

## 1. Introduction

Sudden fluctuations in environmental temperature are one of the most common stressors that all living organisms have to deal with. Cellular membranes are often one of the first indicators of temperature changes, but they can rapidly adapt to new conditions by changing their composition [1,2,3,4]. There are several types of thermoprotectant induced by heat shock (HS), but the family of heat shock proteins (HSPs) and trehalose have been proposed as the main stabilizers of membranes [5,6,7,8]. We previously demonstrated the interplay of membranes, lipids, trehalose, and HSPs in fission yeast during HS [8,9]. In addition to membrane-forming and signaling lipids, storage lipids also play a role in the heat shock response. In fission yeast, the two major storage lipids are triacylglycerols (TG) and ergosteryl esters (EE), which accumulate in lipid droplets (LDs) [10]. The double-knockout yeast mutant (*dga1Δ*/*plh1Δ*; DKO; Table 1), which is unable to produce triacylglycerol, exhibits distinct phenotypes when exposed to mild heat shock. In addition to a prolonged growth arrest, we detected a comprehensive change in the lipidome in response to HS, which differed markedly from those seen in the wild type. We have previously shown that the TG pool can act as a buffer during HS, allowing the cell to rapidly transfer excess unsaturated fatty acids from the membranes as a part of an effective stress management strategy. In the absence of this buffering mechanism, cells can only rearrange their membranes by alternative means, resulting in a suboptimal lipid composition. The differences in the lipidome included elevated signaling lipid generation, such as diacylglycerol (DG) and ceramide, differences in the saturation level of phosphatidylcholine (PC) and phosphatidylethanolamine (PE), defects in the increase of the PC/PE ratio, a higher contribution of medium-chain lipid species to the membrane composition, and reduced lysophospholipid levels. The difference in lipid composition can potentially affect membrane structure and organelle function. The disrupted TG synthesis impairs the adaptability of the membranes to alterations in the environment [11].

LDs are important storage vehicles in lipid metabolism, and thus, they are key players in maintaining the lipid and energy homeostasis of the cells [12,13]. LD biosynthesis is also considered a hallmark of stress across various organisms, including plants and mammals in addition to yeast [14,15,16]. Our group revealed that metabolic crosstalk between membranes and storage lipids facilitates the maintenance of membrane homeostasis in response to HS by accommodating unsaturated fatty acids of structural lipids, enabling their replacement by newly synthesized saturated fatty acids [11]. In the fission yeast, the last steps of TG synthesis are catalyzed by two different enzymes, namely the diacylglycerol O-acyltransferase (Dga1p) and the phospholipid-diacylglycerol acyltransferase (Plh1p) [17,18,19]. Both enzymes use DG as a substrate. Dga1p transfers the fatty acyl chain from acyl-CoA, whereas Plh1p utilizes phospholipids as fatty acid donors to synthesize TG. The precursor DG is known to be one of the curvature-forming lipids and can change the morphology of membranes if present in large enough quantities [20]. There is some evidence that in the absence of the two TG biosynthetic pathways in budding yeast, DG accumulation occurs under glucose starvation, which may result in ‘blob’-like ER fragmentation [21].

The purpose of this study was to investigate whether impaired triacylglycerol metabolism and the resulting perturbation in membrane lipid homeostasis had an effect on the structure and physical state of organellar membranes during HS in fission yeast. The differences in the lipidome of the DKO cells exposed to HS might lead to morphological and biophysical changes in the intracellular membranes and the plasma membrane. To measure the assumed alterations *in vivo*, we employed various fluorescent probes to detect the effect of heat stress on the structure and packaging order of membranes in the DKO cells.
cells-13-01543-t001_Table 1Table 1*Schizosaccharomyces pombe* strains used in this study.NameGenotypeReferenceBRC1h-; leu1-32; ura4-D18[22]BRC48h-; leu1-32; ura4-D18; dga1::ura4; plh1::ura4[11]BRC40h-; leu1-32; ura4-D18; hsp16:GFP(KmR)[8]BRC62h-; leu1-32; ura4-D18; hsp16:GFP(KmR); dga1::ura4; plh1::ura4this studyBRC98h-; pBip1-GFP-AHEL::leu1[23]


## 2. Materials and Methods

### 2.1. Yeast Strains and Growth Conditions

Cells were grown in liquid EMM medium supplemented with leucine and uracil as described [24]. Exponentially growing cells (3–5 × 10^6^ cells/mL) were used for all experiments.

### 2.2. Heat Shock

For all experiments, unless otherwise stated, 20 mL *S. pombe* cultures were heat shocked in a 100 mL Erlenmeyer flask for 1 h at 40 °C in an OLS Aqua Pro Shaking Water Bath (Grant Instruments, Cambridge, UK). Samples were taken at the indicated time points.

For trehalose measurements, 15 mL cultures were filtered (Millipore membrane; 0.45 µm), and the cells were used to inoculate liquid media that were incubated at 40 °C for 20 min before experiments. Additional 15 mL aliquots of cell culture were filtered, and the membranes were placed in preheated (40 °C) medium. After heat shock for 0–60 min, the cells were quickly removed from the medium by membrane filtration, washed with 1× PBS (137 mM NaCl, 2.7 mM KCl, 4.3 mM Na_2_HPO_4_∙7H_2_O, 1.4 mM KH_2_PO_4_, pH 7.3), and frozen in liquid nitrogen for later processing.

### 2.3. Quantitative and Qualitative Analysis of Organelles

Photos were taken with a Zeiss Celldiscoverer 7 microscope using a 50×/1.2 plan-apochromat objective.

For quantifying lipid droplets, multiple methods were tested (ImageJ/FIJI 1.54, CellProfiler 4.2.6), but all of them had problems with accuracy [25]. For our needs, LDs were segmented using Ilastik (version 1.4.0), counted, measured with MATLAB (version 2023a), and summarized with LibreOffice Calc (version 24.8). For other dyes, quantification was performed using ImageJ/FIJI and LibreOffice Calc. All experiments were independently repeated three times.

### 2.4. Generalized Polarization

Aliquots of 100 μL of cell culture from each strain were put into the chamber of our special sample holder (Appendix A). For the control samples without HS, di-4-ANEPPDHQ (Invitrogen, Thermo Fisher, Waltham, MA, USA) was added to each sample to a final concentration of 5 μM [26], and the samples were incubated for 30 min to allow the dye to internalize. Images were taken with a Leica TCS SP5 confocal microscope (Leica Microsystems, Wetzlar, Germany) using a 63× oil immersion objective. For the HS-treated cells, the sample holder was heated up to the desired temperature (40 ± 0.5 °C) with the samples, and then the 1 h long HS was started. At 30 min, the dye was added to the samples to allow its complete internalization by the end of the HS. Images were taken at 40 °C at the end of the HS with the same microscope and settings as described before. For the GP measurement, the emission wavelength ranges 510–590 nm and 610–700 nm were used with excitation at 488 nm. To separate the plasma membrane from the internal membranes, masks were created by the machine learning software Ilastik (version 1.4.0) [27]. From the two-channel data, the GP value for each pixel was calculated using the following formula: GP = (I_510–590_ − I_610–700_)/(I_510–590_ + I_610–700_) (I = pixel intensity). The GP calculation and the false-color image generation were performed with MATLAB (version 2023a).

### 2.5. HSP16 Induction

After HS, 200 μL cultures were used for flow cytometry. From each sample, 50,000 cells were measured with a BD Accuri C6 flow cytometer (Becton, Dickinson and Company, Franklin Lakes, NJ, USA) using CFLow Plus software. The mean FL-1H fluorescence intensity values were used after subtracting the autofluorescence of the cells.

### 2.6. Intracellular Trehalose Levels

Frozen samples were thawed, re-suspended in 100 μL of PBS, and disrupted for 3 min at a speed of 8 in a Bullet Blender (Next Advance Inc., New York, NY, USA) with 0.5 mm zirconium oxide beads at 4 °C. Beads were washed two times with 200 μL PBS, and 100 μL of lysate was boiled for 10 min and centrifuged at 10,000× *g* for 5 min. Trehalase digestion of 25 μL of lysate was carried out in 100 μL of 135 mM citrate buffer (pH 5.6) with 1.15 mU of trehalase (Sigma-Aldrich, Burlington, MA, USA) at 37 °C overnight. Glucose was measured by adding 200 μL of assay reagent (GO Assay kit, Sigma-Aldrich, Burlington, MA, USA), and samples were incubated at 37 °C for 30 min. Reactions were stopped by the addition of 200 μL of 12 N sulfuric acid, and the absorbance at 560 nm was read with a Multiskan EX (Thermo Scientific, Waltham, MA, USA) plate reader. Trehalose and glucose solutions of known concentrations (25–100 μg/mL) were used as standards. Protein concentrations in the samples were measured with the micro-BCA protein assay kit (Thermo Scientific, Waltham, MA, USA, Cat. 23235).

### 2.7. Statistical Analysis

For all significance analyses, unpaired two-tailed Student’s *t*-tests were performed using LibreOffice Calc version 24.8.

## 3. Results

### 3.1. Heat Shock-Induced Changes in LDs in Wild-Type and DKO Cells

To visualize the effect of heat stress on LDs in live cells, a lipid-specific fluorescent probe, LD540 [28], was used in combination with fluorescence microscopy (Table 2). Under normal conditions, a WT cell contains ca. 12 LDs per cell. A one-hour mild heat treatment at 40 °C caused an approximately 20% increase in the number of LDs, ~15/cell (Figure 1A, upper panel, and Figure 1B). In contrast, we could detect only 2 LDs/cell in DKO cells at 30 °C, and the LD count after the HS was 3.4/cell, which was about a 60% increase (Figure 1A, lower panel, and Figure 1B).

The results are in agreement with our previous lipidomics results [11]. In wild-type cells, the significant increase in the number of heat shock-induced LDs is due to elevated levels of TGs. In contrast, in DKO cells, in the absence of TGs, we observed an increase in ergosteryl ester level, which led to the increase in LDs.

### 3.2. Heat Shock Induces Enlargement of Vacuoles in DKO Cells

The yeast vacuoles are dynamic organelles whose structure changes in response to various stresses, including HS. Heat stress can induce negative curvature changes in vacuolar membranes, and it is also the place where certain cell surface proteins are degraded [31]. These vacuoles are similar to mammalian lysosomes, which play a crucial role not only in protein degradation but also in nutrient storage [32]. It has previously been shown that under HS conditions, the size of most organelles in budding yeast was increased [33]. To stain the vacuoles in our experiment, the cells were incubated with the vacuole-specific dye, MDY-64 (Table 2) [29]. The vacuole staining showed a significant difference between the two strains upon HS (Figure 2). Under normal growth conditions (non-HS), we could not detect any difference between the WT and DKO cells (Figure 2A, left panels; Figure 2B–D, 30 °C). HS induced only a slight difference in the size and number of vacuoles in WT cells. A detailed analysis of the images revealed a 7-fold increase in volume and a 50% decrease in the number of vacuoles in DKO cells exposed to HS (Figure 2B–D). Interestingly, in spite of the appearance of oversized vacuoles in the heat-stressed DKO strain (Figure 2A, lower right panel), the size of the cells remained unchanged (12 × 3.5 µm ± 5%).

### 3.3. Heat Stress-Induced Morphological Changes in the Mitochondrial Networks and the Cortical Endoplasmic Reticulum

In addition to the changes in the size and quantity of vacuoles in DKO cells, we investigated the effect of HS on two vital cellular organelles, mitochondria and the cortical ER. The fluorescent probe, MitoTracker Red CMXRos, was used to stain the mitochondrial network (Table 2). The dye enters active mitochondria, and its location overlaps with the known inner membrane protein, cytochrome oxidase subunit I; therefore, it is widely used to monitor mitochondrial activity [34,35]. Inhibition of the respiratory chain/mitochondrial membrane potential drastically decreased the fluorescence intensity of the dye in *S. pombe* [36]. The structure of mitochondrial networks in WT and DKO mutants was very similar at 30 °C (Figure 3A). However, intense blob-like or fragmented structures have been observed in heat-shocked DKO mutants (Figure 3, red arrows), but not in WT cells. The fluorescence intensity of the CMXRos-stained mitochondria proved to be very similar in WT and DKO cells under both normal and HS conditions (Appendix A).

In addition, the mitochondrial network and the morphology of the endoplasmic reticulum were also examined using the ER-specific thermo-yellow probe (Table 2) [30]. To validate the localization of the dye in fission yeast, we have stained the BRC98 strain (Table 1), which carries a cortical ER-targeted GFP [37,38]. As shown in Appendix A, the pattern of the two fluorescent signals overlaps, indicating that ER-thermo-yellow is indeed cortical ER-specific in *S. pombe*. As with the other organelles, there was no observable difference in WT and DKO cells under normal conditions (Figure 3B, 30 °C); however, the ER-thermo-yellow stain revealed a dot-like pattern only in the HS DKO cells (Figure 3B, red arrows).

### 3.4. Highly Altered Membrane Physical State in Heat-Stressed DKO Cells

The impaired lipid remodeling capability we observed previously [11] could lead to the above-described morphological changes of the organelles in heat-shocked DKO cells; therefore, we hypothesized that the failure in membrane homeostasis along with the abnormal organelle structures in the DKO cells could be detected by measuring the lipid packing of the membranes during HS. To determine the packing order of the membranes under different conditions, we used the lipophilic dye di-4-ANEPPDHQ. The dye only fluoresces in a lipid environment, and its emission spectrum depends on the packaging order of the labeled membranes [39]. In order to obtain the most accurate results, we incubated the strains with di-4-ANEPPDHQ in a special, in-house designed, 3D-printed sample holder (Appendix A). After measuring the fluorescence spectra of di-4-ANEPPDHQ in the strains under normal and HS conditions, the generalized polarization (GP) values were calculated. The GP values fell between −1 (more disordered or ‘fluid’ membranes) and +1 (more ordered or ‘rigid’ membranes) [26]. Typical false-color GP images are shown in Figure 4A. There was no difference in the median GP values of either the plasma membrane (PM) or endomembranes (EMs) of the strains in the absence of HS (Figure 4B). The plasma membranes of both strains had a median GP value of ~0.09, while the endomembranes were at 0.06 (Figure 4B), indicating that the EMs were more fluid than the PMs. In contrast, when the strains were exposed to 40 °C, both the plasmalemma and the inner membranes of the DKO mutant were much more disordered than the membranes of the wild-type cells. The median GP values of the heat-shocked WT plasma- and endomembranes were 0.017 and −0.003, respectively, while GP values of DKO membranes were −0.12 for the PM and −0.17 for the EM (Figure 4B).

### 3.5. Impaired Trehalose and HSP16 Production in the Heat-Stressed DKO Mutant

Since the DKO cells’ ability to alter their membranes to adapt to HS is compromised, the synthesis of membrane-associated protectant macromolecules such as trehalose and HSPs was also investigated. First, we checked the heat induction of chromosomally GFP-tagged HSP16p in cells possessing WT and DKO genetic backgrounds, BRC40 and BRC62, respectively (Table 1). At the normal growth temperature of 30 °C, the basal HSP16 level of the DKO cells was significantly lower than that of WT. Also, the heat-treated DKO cells produced 44% less HSP16 compared to the WT cells (Figure 5A). On the other hand, the heat-induced increase in HSP16 expression was significantly higher in the DKO cells (5.4-fold) than in the WT (3.7-fold).

Trehalose is also known to protect membranes under xenobiotic stress [40] and is able to prevent the thermal aggregation of proteins [41]. We followed the trehalose synthesis in the WT and DKO mutant cells during HS at 40 °C, and in the first 20 min of the HS, DKO cells produced ~35% less trehalose than the WT cells (Figure 5B).

## 4. Discussion

Heat is one of the most common stressors that can damage most cellular macromolecules and is especially harmful to membrane homeostasis [7,42,43]. Here we examined the effect of a mutation causing TG deficiency on the plasmalemma and the intracellular membranes of *S. pombe* exposed to 40 °C for 1 h to find out whether the previously demonstrated changes in the lipidome [11] were manifested in the morphology or physical state of the membranes.

First, we determined the lipid droplet (LD) content in WT and DKO cells with or without HS. Not surprisingly, only a few LD540-stained LDs could be detected in DKO cells compared to WT, either under normal or HS conditions (Figure 1A). WT cells had more LDs when exposed to HS, in good agreement with our previously published lipidomic results [11]. In the DKO cells, the few LD540-stained LDs were most likely ergosteryl ester-enriched droplets [44]. The latter possibility is supported by the fact that in the HS cells the ergosteryl ester level significantly increased [11]. One might speculate that in cells that are unable to quickly transfer the unsaturated fatty acids from membrane lipids to triglycerides during HS, ergosteryl ester might partially take on this protective role as a compensatory mechanism [10].

We detected large vacuoles in the DKO but not in the WT cells subjected to HS (Figure 2A). A similar phenomenon has been observed in *Saccharomyces cerevisiae* upon HS, where the area of vacuoles increased parallel with the cell size [33]. In our experiments, the cells’ dimensions remained unchanged (Figure 2A). However, it is well established that signaling might play a key role in vacuole fission and fusion, as these processes are controlled by the mitogen-activated protein kinases (Sty1p, Pmk1p) in fission yeast during osmotic stress [45]. In our previous lipidomic study [11], we found elevated levels of the signaling lipids ceramide and diacylglycerol in the heat-shocked DKO cells, which might facilitate the formation of large vacuoles during thermal stress in the mutant strain. Vacuoles are also in direct contact with other organelles, such as the ER or LDs; therefore, they can affect each other’s organization through contact sites [46,47]. The formation of lipid droplets is also tightly connected to the ER network in fission yeast as well [48].

Since we detected a significant increase in the diacylglycerol level in heat-stressed DKO cells [11], it is reasonable to assume that an excess amount of this curvature-forming lipid contributed to the ‘blob’-like structure of the ER [20,21] and the impaired *de novo* protein synthesis of DKO cells during recovery following heat stress [11]. The mitochondrial network in DKO cells was also affected and showed a similar pattern when cells were exposed to HS (Figure 3A,B). A similar phenotype was observed in acyl-CoA-binding protein (Acb1) mutants showing impaired mitochondrial function and proliferation [49]. However, our results indicate that the fragmented mitochondrial network in heat-stressed DKO cells remains functional (Appendix A). The involvement of diacylglycerol in this process highlights its crucial role, as DG is known to be associated with mitochondrial fusion and fission in Drosophila [50]. In addition, the high amount of DG and ceramide present in the mutant cells contributes to the longer lag phase, which shows strong correlation with the heat-induced damage and recovery of the splicing machinery [51].

Upon determining the membrane packing order by fluorescence microscopy with a specific probe, we found that the plasma membrane was more packed than the endomembranes in both strains under both control and HS conditions. This finding is in good accord with published data [52,53]. It is important to note that changes in sterol levels can affect the fluorescence spectra of the di-4-ANEPPDHQ probe [54]. In *S. pombe*, the plasma membrane contains ergosterol-rich domains, particularly at the cell tips [55,56]. The ergosterol enrichment in these domains and in general in the plasma membrane increases the packing density of the plasma membrane compared to inner membranes, as shown in our false-colored GP images (Figure 4A). Based on our lipidomics data, the observed hyperfluidization of membranes in heat-shocked DKO cells is primarily due to alterations in phospholipid composition [11]. In the absence of TG formation capacity, DKO cells exhibit a reduced ability to effectively and rapidly remove unsaturated fatty acids from their membranes (either directly by Plh1p or indirectly by Dga1p) and replace them with saturated fatty acids, compared to wild-type cells [11]. This impaired stress adaptation ultimately results in excess membrane disordering, as detected by the di-4 ANEPPDHQ probe.

The cellular defense against HS is a complex process with multiple players capable of compensating for each other’s protective effect in their absence. Earlier we showed that HSP16p, a component of the molecular chaperone network, bound to membranes *in vitro*, and its induction was impaired in trehalose-deficient cells exposed to HS at 40 °C [8]. In the DKO strain, we also observed a lower HSP16p level in cells incubated at 40 °C, although it displayed a higher induction rate compared to WT. The reduced HSP16p level can be explained by the impairment of *de novo* protein synthesis [11] and the damaged structure of the cortical ER (Figure 3B). However, the reason for the lower baseline level of HSP16p in DKO cells has yet to be determined. In fission yeast cells exposed to HS, trehalose is synthesized relatively rapidly (Figure 5B). The lower HS-induced trehalose level in DKO cells could not be explained by the deficiency in *de novo* protein synthesis because inhibition of translation had no effect on the trehalose level during HS [57]. Therefore, we assume that other processes, such as impaired perception, signaling, or regulation, could be responsible for this phenomenon.

Taken together, our findings provide data that triacylglycerol metabolism contributes to cellular defense by mitigating the detrimental effects of heat shock on membranes. In support of our hypothesis, we demonstrated that the inner membranes of TG-deficient cells displayed altered architecture with fewer LDs, larger vacuoles, and dot-like stained ER and mitochondrial structures after HS. In addition, we proved that DKO cells were unable to normally regulate the physical state of both the plasma membrane and the inner membranes when subjected to even mild HS. We also demonstrated a new tool to visualize the structure of the cortical ER in *S. pombe* and a readily available, cost-effective solution for the frequently encountered technical problem of proper temperature control in confocal microscopy.

## Figures and Tables

**Figure 1 cells-13-01543-f001:**
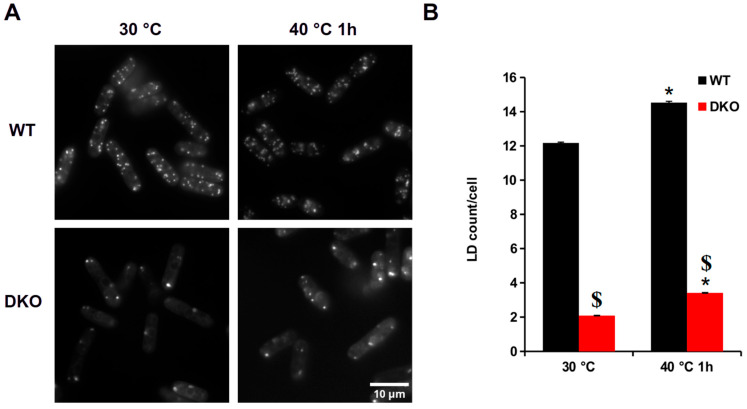
Analysis of lipid droplets in WT and DKO cells subjected to HS. (**A**) LD540 staining of LDs in WT and DKO cells before and after HS. (**B**) Quantitative analysis of the LDs in the WT and mutant cells. Data are mean ± SEM of *n* = 3 independent experiments with ≥500 cells/experiment; * *p* < 0.05, 30 °C vs. 40 °C, 1 h; $ *p* < 0.05 WT vs. DKO.

**Figure 2 cells-13-01543-f002:**
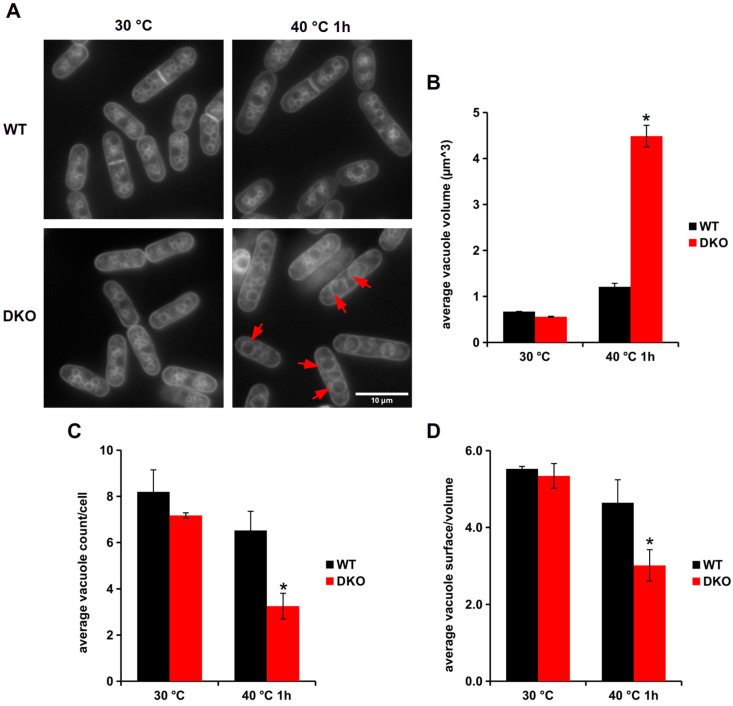
Analysis of vacuoles of WT and DKO exposed to HS. (**A**) Representative images of MDY-64-stained WT and DKO cells. Red arrows indicate enlarged vacuoles in heat-shocked DKO cells. Analysis of the vacuolar size (**B**), quantity (**C**), and surface/volume (**D**) after HS in both strains. Data are mean ± SEM of *n* = 3 independent experiments with >200 vacuoles analyzed; * *p* < 0.05, 30 °C vs. 40 °C 1 h.

**Figure 3 cells-13-01543-f003:**
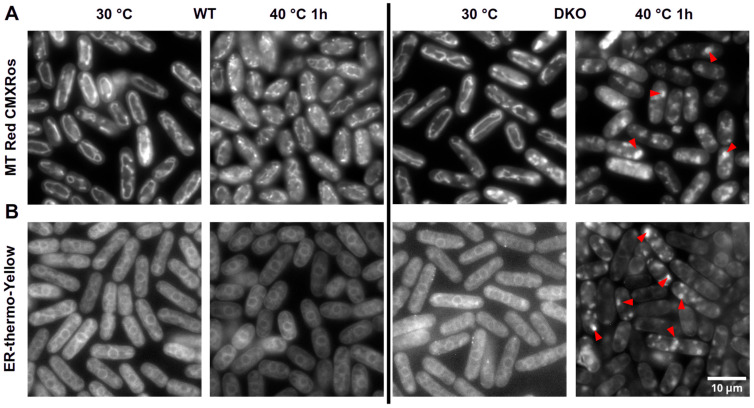
Changes in the mitochondrial network and cortical ER of heat-stressed WT and DKO *S. pombe* cells. (**A**) MitoTracker CMXRos staining; disordered parts of the mitochondrial network are indicated by arrows. (**B**) ER-thermo-yellow staining; dot-like ER structures are indicated by red arrows.

**Figure 4 cells-13-01543-f004:**
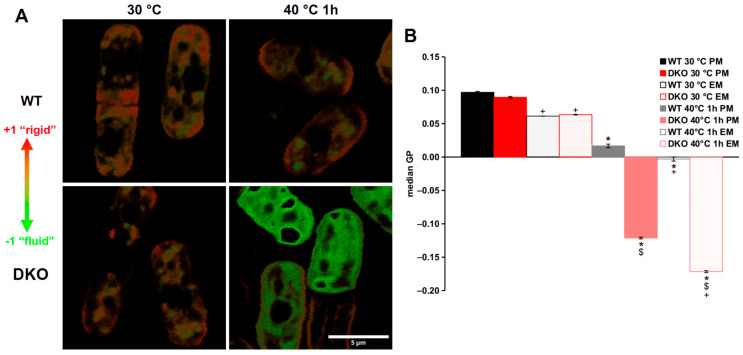
Assessment of lipid packing of membranes in the wild-type and DKO strains. (**A**) False colored GP images of the WT (upper panel) and DKO (lower panel) cells during heat shock. (**B**) Median GP values of plasma membranes (PM) and endomembranes (EM) of control and heat-shocked WT and DKO cells. Data are mean ± SEM of *n* = 4 independent experiments, >90 cells per data point analyzed; * *p* < 0.05 30 °C vs. 40 °C, 1 h; ^$^ *p* < 0.05 WT vs. DKO; ^+^ *p* < 0.05 PM vs. EM. GP, generalized polarization.

**Figure 5 cells-13-01543-f005:**
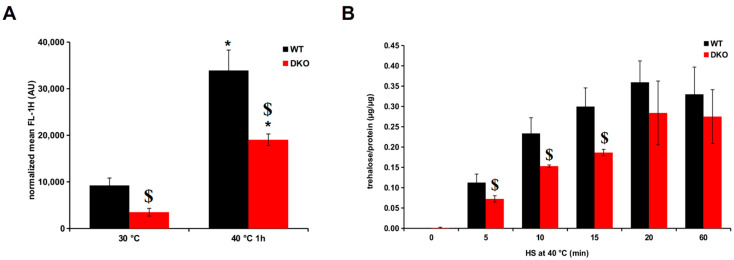
Comparison of the expression of the thermoprotectants HSP16 and trehalose in heat-stressed WT and DKO cells. (**A**) Induction of HSP16-GFP in the WT (BRC40) and DKO (BRC62) backgrounds. (**B**) Accumulation of trehalose in WT and DKO cells during heat treatment. Data are mean ± SD for n = 3 independent experiments; * *p* < 0.05 at 30 °C vs. 40 °C for 1 h; ^$^ *p* < 0.05 in WT vs. DKO.

**Table 2 cells-13-01543-t002:** Conditions for staining organelles after heat shock.

Organelle	Dye	Final Conc	Incubation, min	Reference
lipid droplets	LD540	50 ng/mL	15 min	Limes-Institut-Bonn [28]
vacuoles	MDY-64	10 µM	15 min	Invitrogen, Thermo Fisher [29]
cortical ER	ER-thermo-Yellow	1 µM	25 min	[30]
mitochondria	MitoTracker Red CMXRos	350 nM	15 min	Invitrogen, Thermo Fisher

## Data Availability

The original contributions presented in this study are included in the article/Appendix A; further inquiries can be directed to the corresponding author.

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
