# Peer review of "Mild Heat Stress Alters the Physical State and Structure of Membranes in Triacylglycerol-Deficient Fission Yeast, Schizosaccharomyces pombe"

_cells, 2024, doi:10.3390/cells13181543_

Round 1

Reviewer 1 Report

Comments and Suggestions for Authors

In their original research article Péter Gudmann and colleagues study how the deficiency in triacylglycerol affects cell organelles organization and plasma membrane and inner membranes order in Schizosaccharomyces pombe. The Authors are able to show that upon heat shock treatment the deficiency in triacylglycerol promotes changes in lipid droplets amount, vacuole volume and amount, mitochondrial and endoplasmic reticulum architecture. The manuscript is well written and organized. Prior to publishing, however, the Authors should address several concerns in order to improve the impact and consistency of their work.

Major concerns:

-        In the introduction it is important to briefly explore the lipidome changes they mentioned both in the introduction and Discussion.

-        The Authors do not mention which statistical test they are using to evaluate the statistical significance of the differences they found. In particular, I believe that a two-way Anova should have been used to analyse the data in Figs. 1, 2, 4 and 5.

-        The authors are silent on how the vacuolar volume (and surface-to-volume ratio) was calculated.

-        Regarding vacuole analysis:

o   for a more thorough result analysis, I suggest the Authors to also determine the total vacuolar volume per cell;

o   considering the info in lines 159-160 (“Interestingly, in spite of the appearance of oversized vacuoles in the heat-stressed DKO cells (Fig. 2A, lower right panel), their size remained unchanged (12 × 3.5 μm ± 5%)”), I suggest the Authors to plot the size of vacuoles in WT and DKO cells at 30ºC and after HS. And the size of vacuoles in heat-stressed DKO remains unchanged compared to which condition? DKO 30ºC? WT 40ºC? Nonetheless, I find it difficult to visualize how the vacuoles’ size remains unchanged considering that in HS DKO cells there is a great increase in vacuolar volume concomitant to a decrease in the amount of vacuoles per cell. Moreover, the surface-to-volume ratio is also decreasing in HS DKO, which is indicative of bigger vacuoles.              

-        I do not fully agree with how the Authors present their results in lines 132-135: “A one-hour mild heat treatment at 40°C caused an approximately 20% increase in the number of LDs, ~15/cell (Fig. 1A, upper panel, and 1B). In contrast, we could detect only 2 LDs/cell in DKO cells at 30°C, and the heat-induced LD increase was only 20% of that observed in WT cells (Fig. 1A, lower panel, and 1B).” For a more coherent result description the % increase in the number of LDs for both strains should presented. As the Authors mentioned, in WT cells there is a 20% increase in the number of LDs (from ~12 to almost 15), but in DKO cells the increase in the number of LDs is certainly over 50% since they go from ~2 to over 3. So, in fact, the % increase in LDs is higher DKO cells. On the other hand, the absolute increase in the LDs number in DKO cells is ~1.5, which is 60%, not 20%, of the increase in the LDs number in WT cells (~2.5).

-        A major concern is the fact that the Authors used di-4-ANEPPDHQ to assess variations in the lipid order of plasma membrane and endomembranes. Although di-4-ANEPPDHQ may be employed to evaluate membrane order, it is not the most suitable probe for that purpose as pointed out in the work by Mariana Amaro, Francesco Reina, Martin Hof, Christian Eggeling and Erdinc Sezgin (Laurdan and Di-4-ANEPPDHQ probe different properties of the membrane, J Phys D Appl Phys. 2017 Apr 5; 50(13): 134004. https://www.ncbi.nlm.nih.gov/pmc/articles/PMC5802044/). To be brief they state in their conclusions: “…On the other hand, the results for di-4-ANEPPDHQ dye revealed complex relaxation kinetics involving multiple processes. The GPdi-4 values do not correlate with lipid packing and are influenced by cholesterol in a specific way. This discrepancy may result in several factors including interactions between the dye and other membrane components or the exact location of the dye in the membrane [55]. It is of particular importance that di-4-ANEPPDHQ is an electrochromic dye, i.e. its fluorescence emission spectrum is sensitive to the membrane potential. For example, the transmembrane potential of the plasma membrane ranges from around –40 mV to –80 mV, while that of the mitochondrial membrane is around  140 to  180 mV. Consequently, the electrochromic property of di-4-ANEPPDHQ can substantially bias the interpretation of empiric values gathered from cell biology experiments. Therefore, GPdi-4 seems not to be the best indicator of lipid membrane order.”

-        Since di-4-ANEPPDHQ GP values can be strongly influenced by the sterol levels, Laurdan would be the most suitable choice if the purpose is to assess membrane order.

-        The make Their work more consistent the Authors should perform these experiments with Laurdan. Moreover, when discussing the results obtained for di-4-ANEPPDHQ the Authors must address the impact that sterol and, consequently, membrane potential may have in GP values and cite the work I mentioned above.

-        Lines 234-235: “Also, the heat-treated DKO cells produced 44% less HSP16 compared to the WT cells…” – However DKO cells increase their HSP16 levels almost 7x, whereas WT HSP16 levels increase a little less than 4x.

-        After following trehalose synthesis in the WT and DKO mutant cells during HS at 40°C, the Authors state that “at all measured time-points, DKO cells produced 20-35% less trehalose than the WT cells”. I do not completely agree since above 20 min the error bars are so large that the trehalose values in WT and DKO cells are not statistically significant.

Minor concerns:

-        Line 69: “growing cells (3–5×106 cells/mL)” – 3–5×106 cells/mL

-        Fig. S2 is mentioned in text prior to Fig. S1.

-        In line 218 the Authors mention “The median GP values of the heat-shocked WT plasma- and 218 endomembranes were close to ~0”. I suggest the Authors to make explicit the GP values as They do for the other situations and also because the GP value of WT plasma membrane is close to 0.016.

-        Line 231: “HSPs were also investigated” – “was” instead of “were”.

Comments on the Quality of English Language

English quality is good. Only minor editing.

Reviewer 2 Report

Comments and Suggestions for Authors

Gudmann et al determined the effect of triglyceride (TG) deficiency, induced by knockout of 2 enzymes required for TG synthesis, on the heat stress response of S. pombe. They concluded that TG deficiency greatly increased the detrimental effects of heat stress on cellular functions and intracellular organelles.

The manuscript is well written and the conclusions are supported by the data.

Since TGs are not components of cellular membranes but segregate into lipid droplets, it is not intuitive why TG deficiency has such strong effect on intracellular membranes. Therefore, some speculations about the mechanism by which TG deficiency affects the heat shock response would make the paper more interesting.

Round 2

Reviewer 1 Report

Comments and Suggestions for Authors

The Authors addressed all my concerns and the manuscript can be published as it is.